# Polypeptide formation in clusters of $\beta$-alanine amino acids by single ion impact

Patrick Rousseau [1✉], Dariusz G. Piekarski [2], Michael Capron[1], Alicja Domaracka[1], Lamri Adoui [1], Fernando Martín [2,3,4], Manuel Alcamí [2,4,5], Sergio Díaz-Tendero [2,3,5✉] & Bernd A. Huber[1]

The formation of peptide bonds by energetic processing of amino acids is an important step towards the formation of biologically relevant molecules. As amino acids are present in space, scenarios have been developed to identify the roots of life on Earth, either by processes occurring in outer space or on Earth itself. We study the formation of peptide bonds in single collisions of low-energy $He^{2+}$ ions ($\alpha$-particles) with loosely bound clusters of $\beta$-alanine molecules at impact energies typical for solar wind. Experimental fragmentation mass spectra produced by collisions are compared with results of molecular dynamics simulations and an exhaustive exploration of potential energy surfaces. We show that peptide bonds are efficiently formed by water molecule emission, leading to the formation of up to tetrapeptide. The present results show that a plausible route to polypeptides formation in space is the collision of energetic ions with small clusters of amino acids.

[1] Normandie Univ, ENSICAEN, UNICAEN, CEA, CNRS, CIMAP, 14000 Caen, France. [2] Departamento de Química, Módulo 13, Universidad Autónoma de Madrid, 28049 Madrid, Spain. [3] Condensed Matter Physics Center (IFIMAC), Universidad Autónoma de Madrid, 28049 Madrid, Spain. [4] Instituto Madrileño de Estudios Avanzados en Nanociencias (IMDEA-Nanociencia), Cantoblanco, 28049 Madrid, Spain. [5] Institute for Advanced Research in Chemical Sciences (IAdChem), Universidad Autónoma de Madrid, 28049 Madrid, Spain. ✉email: patrick.rousseau@unicaen.fr; sergio.diaztendero@uam.es

One of nature's most important classes of organic polymers is formed by peptides and proteins, which are linear chains, usually built up from a construction kit of 20 canonical $\alpha$-amino acids. They are linked by a covalent N–C amide (or peptide) bond connecting the carboxyl group of one amino acid with the amino group of the subsequent one. The formation of a peptide bond implies the release of a single water molecule. Peptides and proteins are naturally formed in living cells, but they can also be artificially produced in chemistry laboratories in liquid and solid phases[1–3].

A number of scenarios towards abiotic peptide synthesis focuses on the idea that simple amino acids came down on Earth or exoplanets from outer space[4]. It is well established that various astrophysical environments contain small biologically relevant molecules[5] whose signatures were detected, e.g., in the interstellar medium by infrared and microwave spectroscopy[6]. Further evidence comes from the mass analysis of samples from carbonaceous chondrite meteorites[7], which are believed to have originated from interstellar dust particles[8]. Here, amino acids (among them $\beta$-alanine, glycine, and $\alpha$-alanine being the most abundant ones[9]) can for instance be formed upon UV photon impact on ice grains[10–12] or by UV-activated gas-phase chemistry[13,14].

Our work aims to answer a more general and comprehensive question: are there direct gas-phase routes from neutral, cold gas-phase ensembles of amino acids toward formation of polypeptides? Recently, several theoretical and experimental studies have been performed in order to answer this question. Amino acid clusters (dimer and tetramer clusters of $\alpha$-alanine) have been studied by theory as precursors for polypeptide formation as the orientation of the OH $\cdots$ NH mode of hydrogen bonding is found to be suitable for chemical condensation[15]. In the case of two glycine molecules[16] or double amino acids[17], different mechanisms for peptide bond formation have been mapped out with quantum mechanical electronic structure methods. Furthermore, peptide bond formation has also been suggested to be initiated by glycine protonated at the hydroxyl oxygen[18]. The process is found to be barrier-free and clearly exothermic, thus, peptide bond formation could take place even under interstellar conditions.

Experimental work has concentrated on amide bond formation in the context of peptide synthesis[19,20] or the application of electrospray high-resolution mass spectrometry[21]. In the first case, controlled peptide extensions (N- or C-terminal) in catalyzed reactions are discussed. In the second case, it is shown that collision-induced dissociation is efficient to form peptides from dimer or larger systems. Also, collisions between photons (VUV, 157 nm) and proton-bound peptide complexes, including protonated dimers, have been used to show the formation of longer amino acid chains by water elimination[22]. This technique allowed for the linear coupling of amino acid chains to produce oligopeptides from peptide complexes[23,24].

Collisions of ions with large complex molecules and clusters can also be an efficient tool to induce chemical reactions. This is due to the transfer of energy, momentum, and charge during the collision, which allows the system to overcome existing reaction barriers or to create highly reactive species that interact immediately with surrounding cluster constituents. In this context, it has been shown that coalescence and growth reactions occur when highly charged Xe ions collide at 500 keV with van der Waals clusters of fullerene molecules, forming a nanoplasma, resulting finally in the formation of giant fullerenes[25]. Another example concerns collisions involving slow heavy projectiles with kinetic energies of a few keV in very low charge states. In this case, elastic nuclear collisions dominate the interaction, leading to the formation of reactive fragments by knockout processes that can react with neighboring molecules. Thus, the molecular growth of pyrene molecules could be demonstrated in low-energy

ion collisions with pyrene clusters yielding a variety of new, much larger molecules[26]. Or, in van der Waals clusters of fullerene molecules, grain particles could be produced containing up to ~1400 covalently bound carbon atoms[27].

In this work, we show the formation of peptide bonds in collisions of slow He$^{2+}$ ions with $\beta$-alanine clusters. The mechanisms responsible for this process are analyzed with the help of state-of-the-art quantum chemistry calculations. At variance with polymerization reactions resulting from collisions with heavy ions, knockout processes are less important due to the low He mass, so that electronic excitation and ionization processes are expected to be dominant.

## Results

**Fragmentation mass spectra resulting from ion collisions**. We have studied the interaction of $\alpha$-particles (He$^{2+}$ ions) with kinetic energies that are typical for solar wind ions with cold, hydrogen-bound neutral clusters of $\beta$-alanine molecules ($\beta$-ala, NH$_2$CH$_2$CH$_2$COOH, mass 89 u). In the present collision energy region, the ionization of the target occurs mainly by single- and double-electron capture which in both cases is an exothermic process. The cationic products are analyzed by time-of-flight mass spectrometry. The intensity of the produced singly charged clusters decreases with increasing cluster size. Additional cluster fragments are observed between the integer cluster numbers ($\beta$-ala)$_n$. Details on the size range between the monomer and the trimer are shown in Fig. 1a, b.

It turns out that the dominant singly charged clusters are protonated $[(\beta - \text{ala})_n \text{H}]^+$, being characterized by the $m/z$ ratios 90, 179, and 268 (see Fig. 1). The observed protonation is in very good agreement with the results obtained in the molecular dynamics (MD) calculations, which have been performed using the density functional theory (DFT-based MD) for different cluster sizes, charges, and internal energies (see details in the Supplementary Information). The simulations show that in excited and ionized clusters of $\beta$-alanine molecules, intermolecular proton transfer takes place with high efficiency leading to singly charged protonated dimers and trimers. Therefore, the protonated clusters observed experimentally are formed by ion-induced chemical reactions in larger ionized clusters. In all the cases, the protonation occurs at the more basic center, i.e., the amino group.

In addition to the protonated monomer, the mass spectra show protonated dimer and trimer ions, which are always due to fragmentation of larger clusters, and intense contributions at masses of $m/z = 134, 144, 161, 223, 232$, and $250$. Their relative intensity increases with initial cluster size, which is particularly visible for the $m/z = 161$ products. The identification of the important fragments, experimentally observed between the monomer and the trimer clusters, is given in Supplementary Table 2. The ion masses and charge states as well as their structures, calculated with DFT, are specified. As can be seen in Fig. 1, several fragments are produced by the loss of a fragment with the mass 18, i.e., the loss of one water molecule, leading to the production of the fragments with $m/z = 161$ ($179 \rightarrow 161 + \text{H}_2\text{O}$) and $m/z = 250$ ($268 \rightarrow 250 + \text{H}_2\text{O}$), or the loss of 2 water molecules yielding $m/z = 232$ ($268 \rightarrow 232 + 2\,\text{H}_2\text{O}$). These fragments demonstrate the formation of peptide bonds and the production of the corresponding di- and tripeptide systems. This is confirmed by the MD simulations (see Supplementary Fig. 3) and the exploration of the potential energy surface, as described in the following section.

## Discussion

In our theoretical approach, we are implicitly assuming that the cations generated in the collision are in the ground electronic

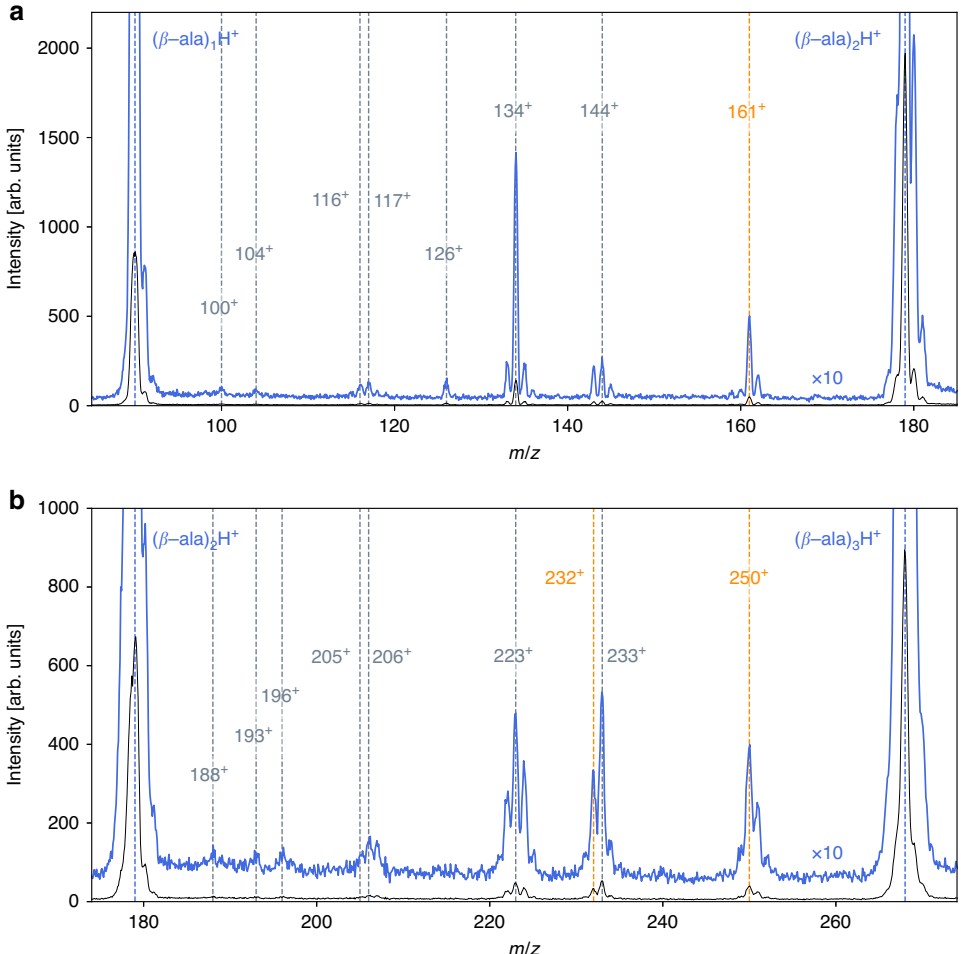

**Fig. 1 Fragmentation mass spectra.** Singly ionized reaction products of collisions between 30 keV $He^{2+}$ projectiles and cold neutral $\beta$-alanine clusters. Blue curves correspond to a 10× enhancement of the intensity. **a** Region between monomer and dimer; **b** region between dimer and trimer. The numbers indicate the mass-over-charge ratio ($m/z$) of different fragments and products. Peptide species produced after the collision are indicated in orange, while other covalent products are in gray.

state. This is based on the fact that (i) the collision is much faster than fragmentation (the typical collision time is of the order of the femtosecond) and (ii) the energy available in the excited electronic states is rapidly redistributed into the nuclear degrees of freedom due to either the very efficient non-adiabatic couplings (e.g., through conical intersections) between these electronic states or the very dense manifold of vibrational states associated with such large systems. In this way, one can reasonably expect that the initial excitation energy has already been transferred to the nuclei (in the form of nuclear velocities) when fragmentation starts. This approach has been shown to accurately describe fragmentation dynamics in similar experiments in the past, see, e.g., refs. [28–32].

We have explored the potential energy surfaces (PES) of several singly and doubly charged dimer and trimer systems in order to clarify the experimentally observed fragments. In the molecular dynamics simulations, doubly charged species are also produced for larger initial cluster sizes; thus we also consider them in our study of the PES. The energy levels in the diagrams presented here are referred to the most stable neutral dimer/trimer structures (previously studied in ref. [33]). As can be seen, relative energies obtained in DFT and coupled cluster theory (CCSD) are very similar, which gives strong support to the discussion based on the calculated PES (see Supplementary Information for more details about the comparison between these two sets of calculations).

As a general trend, we observe that the protonated structures lie at a lower energy than the nonprotonated ones, showing a strong proton affinity and thus a large stabilization after protonation. Most probably, this is the driving force that explains the experimental observations. In the collision, charge and energy are transferred to a large weakly bound cluster, leading to emission of neutral monomers and other moieties, accompanied by intracluster proton transfer, which thus stabilizes smaller protonated structures. These ones evolve toward the production of the experimentally detected fragments as detailed in the pathways presented below. Proton transfer facilitates reactivity since the energy of the protonated dimer lies below that of the most stable neutral structure (see structure $[(\beta - ala)_2 + H]^+$ in the PES shown in Fig. 2), as well as below the first ionization threshold (~9.7 eV). That is, proton transfer stabilizes the singly charged dimer, and there is no need for extra energy to make a peptide bond. In contrast, the analysis of the potential energy surface for the ionized, nonprotonated dimer, $[(\beta - ala)_2]^+$, shows that covalent bond formation requires additional energy, as the structure of the ionized dimer lies ~9 eV above the energy position of the neutral dimer while those for the covalent products lie at 10.5 eV and 17.7 eV and does not involve water release (see Supplementary Fig. 5). The extra energy is taken from the exothermicity of the electron-capture process. Proton transfer also stabilizes the doubly

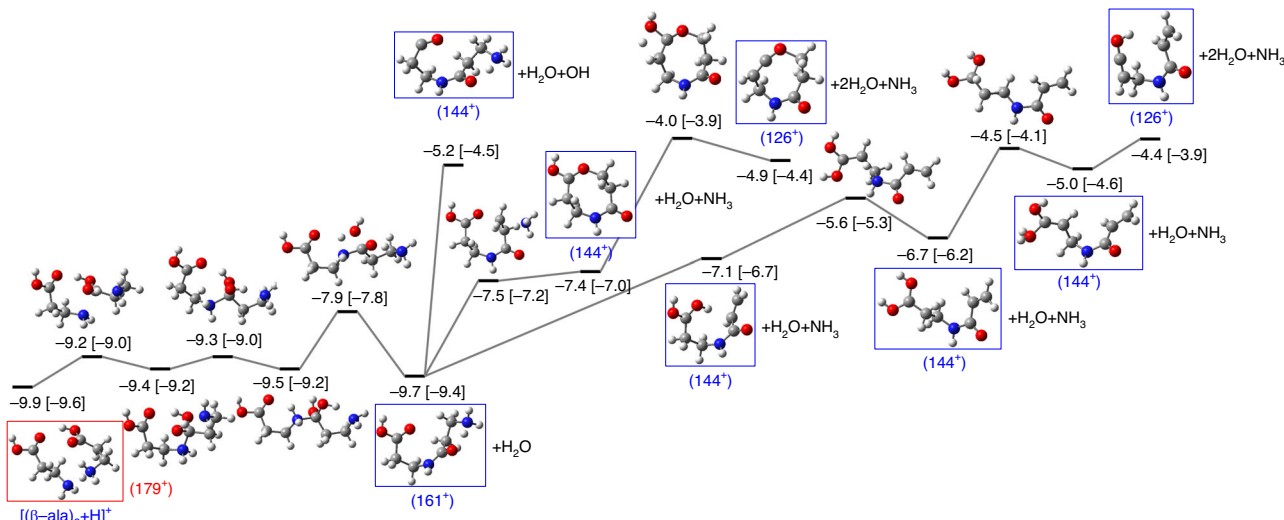

**Fig. 2 Potential energy surface exploration of the singly charged protonated dimer.** Fragmentation pathways starting from the singly charged protonated dimer ($m/z = 179$) showing the formation of the dipeptide ($m/z = 161$, noted 161$^+$) and fragments ($m/z = 144$) and ($m/z = 126$), respectively noted 144$^+$ and 126$^+$. Red box: initial configuration; blue boxes: observed fragments. Notice that the 1st vertical ionization potential for the neutral dimer is 9.7 eV, i.e., after proton transfer the dimer is highly stabilized. Geometries optimized at the DFT-M062X/6-311++G(d,p) level of theory. Relative energies (in eV) computed at the CCSD/6-311++G(d,p) level over the geometry previously obtained and referred to the most stable neutral dimer are given next the molecular structures. In square brackets: relative energy computed with DFT-M062X/6-311++G(d,p). H atoms are given in white, C atoms in gray, N atoms in blue, and O atoms in red.

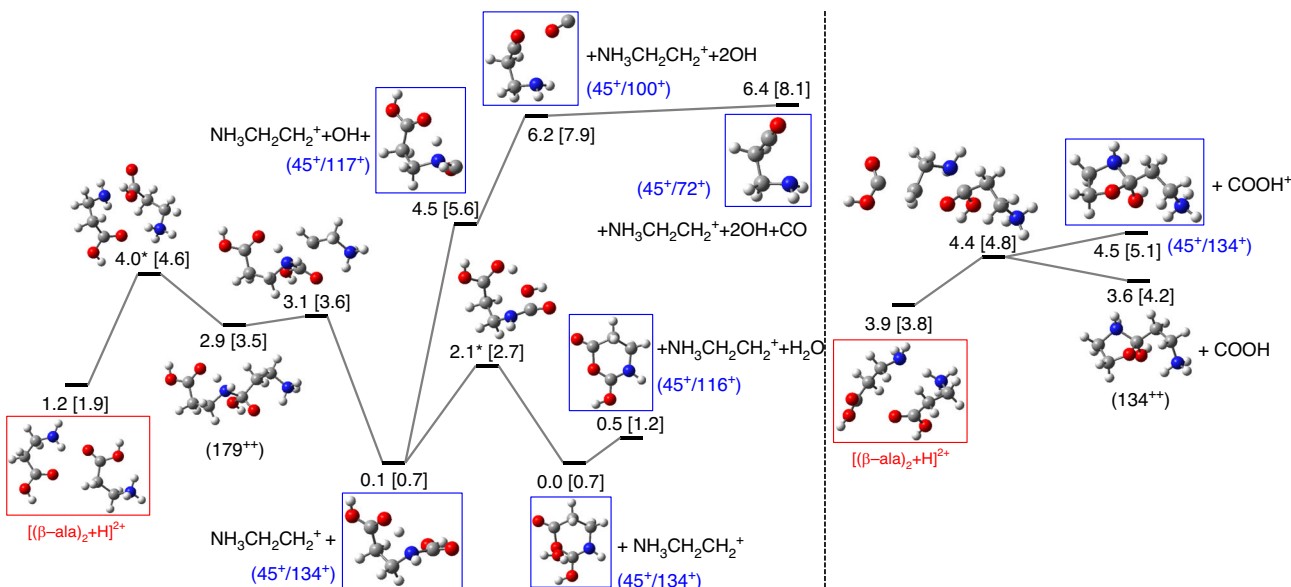

**Fig. 3 Potential energy surface exploration of the doubly charged protonated dimer.** Fragmentation pathways starting from the doubly charged protonated dimer ($m/z = 179/2$) leading to ion pair formation, relating $m/z = 45$ with $m/z = 134, 117, 116, 100,$ and 72. Red box: initial configuration; blue boxes: observed fragments in coincidence. Notice that the 1st vertical ionization potential for the neutral dimer is 9.7 eV and the 1st + 2nd vertical ionization potential is 27.4 eV. Geometries optimized at the DFT-M062X/6-311++G(d,p) level of theory (structures with an asterisk were computed at the AM1 level). Relative energies (in eV) computed at the CCSD/6-311++G(d,p) level over the geometry previously obtained and referred to the most stable neutral dimer are given next to the molecular structures. In square brackets: relative energy computed with DFT-M062X/6-311++G(d,p). H atoms are given in white, C atoms in gray, N atoms in blue, and O atoms in red.

ionized dimer, see structure $[(\beta - \mathrm{ala})_2 + \mathrm{H}]^{2+}$ in Fig. 3. The pathways starting from this structure lie below the ionization threshold, and the corresponding reactions lead to products that are observed in the experiments. This analysis gives additional support to the idea that the ionizing collision induces intracluster proton transfer followed by the release of the excess energy through evaporation of neutral moieties.

In Fig. 2, we show the part of the potential energy surface describing the fragmentation of the weakly bound dimer ($m/z = 179$), which is protonated in one of the amino groups $[(\beta - \mathrm{ala})_2 + \mathrm{H}]^+$ (red box; parallel molecular orientation of both molecules; relative energy $-9.9$ eV with respect to the neutral dimer). In a first step, the proton is transferred to the C=O group forming a diol structure (structure at $-9.4$ eV), in which the new

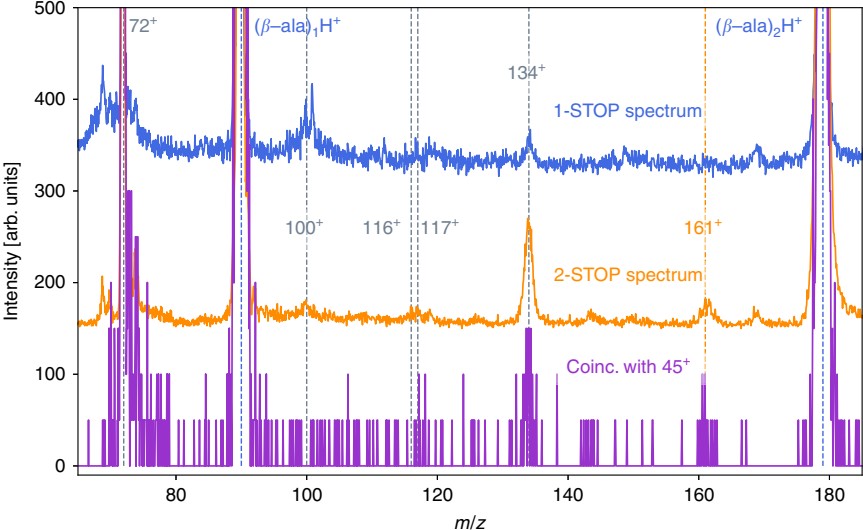

**Fig. 4 Coincidence mass spectra.** The top spectrum (blue curve) corresponds to the one-stop spectrum containing events producing one single-charged fragment while the middle spectrum (orange curve) corresponds to the two-stop spectrum containing all events producing two charged fragments. The bottom spectrum (purple curve) corresponds to those fragments which are measured in correlation with the fragment $m/z = 45$. Peptide species produced after the collision are indicated in orange, while other covalent products are in gray.

bond between C and N is already formed. However, this is not a peptide bond as a water molecule has not been lost yet. In a second step, one of the OH group of the diol gets close to the terminal $NH_2$ (structure at $-9.5$ eV) followed by a concerted mechanism[18], where one H is transferred from OH to the terminal $NH_2$ and a water molecule is lost, leading to the formation of a protonated dipeptide with $m/z = 161$. This step has to overcome a high energy barrier of 1.6 eV. Molecular dynamics simulations confirm that water release is indeed accompanied by the formation of the peptide bond (see details in the Supplementary Information).

In Fig. 2, we also show that when the internal energy is sufficiently high, the experimentally observed fragments with $m/z = 144$ and $m/z = 126$ can be formed. According to the simulations, these peaks correspond to: first the loss of one $H_2O$ molecule (forming a peptide bond through a stepwise mechanism involving a diol intermediate, as discussed before), second the loss of $NH_3$ or OH giving the fragment peak at $m/z = 144$, and finally the loss of a second $H_2O$ molecule leading to fragments with $m/z = 126$. Both fragments at $m/z = 144$ and 126 can be formed in open or in closed ring structures, the latter ones being more stable.

Further peaks are observed in Fig. 1 with mass/charge values of 100, 104, 116, 117, and 134, which have not been discussed before. Therefore, we show in Fig. 3, reaction pathways of the doubly charged protonated dimer ($m/z = 179/2$, energetically positioned at 1.2 eV with respect to the neutral dimer) are shown. This situation would reflect those cases where double ionization and protron transfer take place, leading to a protonated and further ionized cluster $[(\beta - ala)_n H]^{2+}$. We detail the different mechanisms that explain the experimental observation of five different fragments, some of them not observed in the $[(\beta - ala)_2 H]^+$ case. We have found different fragmentation pathways for the two relative orientations of the $\beta$-alanine molecules in the initial configuration: The parallel orientation (right part of Fig. 3) leads to the emission of the COOH unit in its neutral form, leaving behind the doubly charged fragment $m/z = 134/2$ at an energy level of 3.6 eV. The emission of the charged fragment $COOH^+$ forming the ion pair (45+/134+) may occur at the higher energy of 4.5 eV. The ion pair (45+/134+) is also formed from the antiparallel orientation of both molecules

following the mechanism, shown in the left part of Fig. 3. In this case, the fragment $NH_3CH_2CH_2^+$ ($m/z = 45$) is first emitted leading to the intermediate state at 0.1 eV. This path is followed by the loss of an $H_2O$ molecule through an energetic barrier of 2 eV, forming the stable ring structure to which a water molecule is weakly attached. Emission of this water molecule leads to the fragment $m/z = 116$. Thus, the ion pair (45+/116+) is formed at ~0.5 eV. Formation of the fragment $NH_3CH_2CH_2^+$, can also be followed by subsequent losses of neutral OH and CO molecules, and is related to the peaks $m/z = 117$, 100, and 72 in correlation with $m/z = 45$. Although these paths require much higher internal energies, most of the predicted ion pairs are associated with the singly charged molecule $NH_3CH_2CH_2^+$ ($m/z = 45$). Notice that, although all these pathways appear at higher energy than the neutral dimer, the barriers are always below 10 eV. In ion-molecule/cluster experiments, as the one shown here, excitation energy of a few eVs are expected[31,34].

Information on the correlation between positively charged fragments, being formed within one collision event, can be obtained experimentally from the analysis of so-called $k$-stop mass spectra. These spectra correspond to collision events where exactly $k$-charged fragments are detected in a single-collision event. This implies, that the fragmenting parent system, formed in the collision, has to be at least $k$ times charged (for more details see the section Experiment and the Supplementary Information). Therefore, we compare in Fig. 4 the spectrum for events characterized by one stop (mostly singly charged parent clusters) and that for two stops (mostly doubly charged parent clusters). Noting that in the latter case, the internal energy of the clusters is expected to be larger than in the case of single ionization, which occurs at larger interaction distances. In addition, we show the ion distribution for fragments which are observed in correlation with the fragment $NH_3CH_2CH_2^+$ ($m/z = 45$). On the one hand, we observe that in the single-stop spectrum the intensity of the dipeptide ($m/z = 161$) is very weak, if not negligible. In the two-stop spectrum, we observe a strongly enhanced intensity for the dipeptide. This can be interpreted as the decay of initially doubly charged clusters into protonated dimers (see Supplementary Table 1), which can be followed by the formation of dipeptides according to the pathway shown in Fig. 2. On the other hand, the

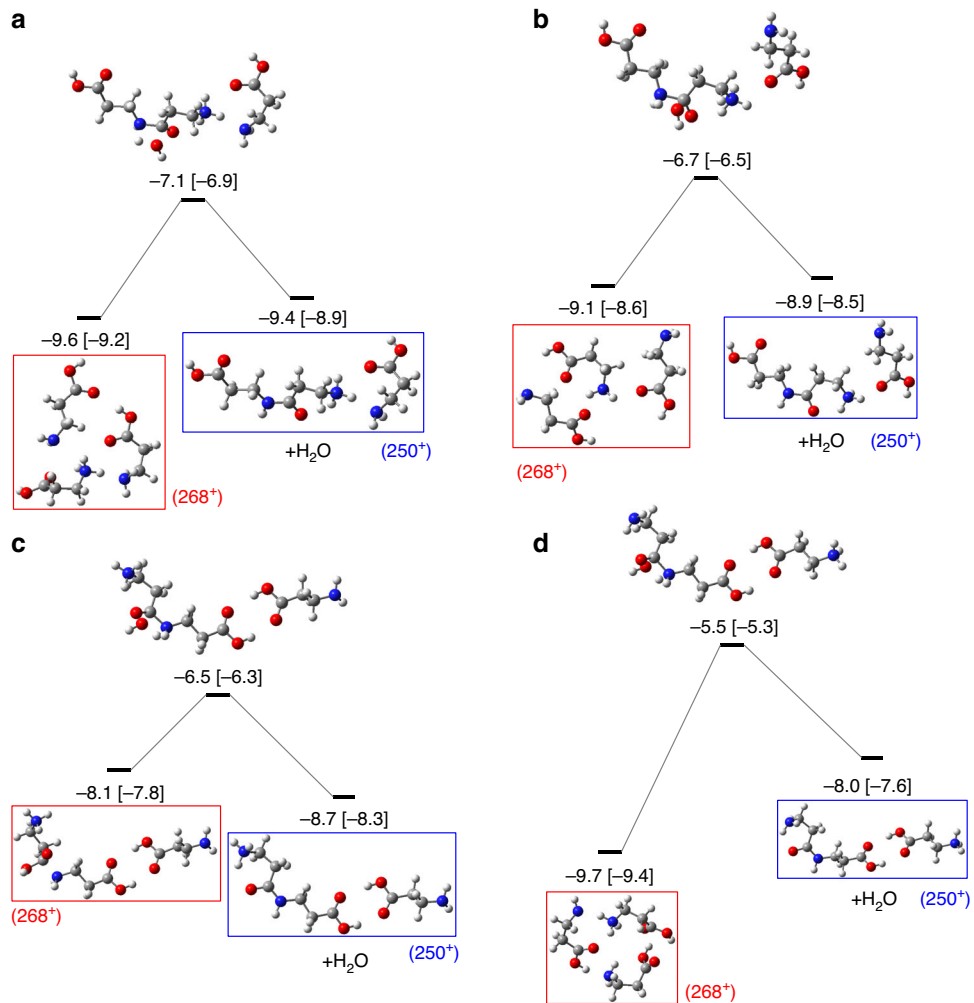

**Fig. 5 Potential energy surface exploration of the singly charged protonated trimer.** Determining steps of dipeptide formation starting from different initial N-protonated trimer systems ($m/z = 268$), shown in red boxes, and leading to a N-protonated dipeptide with a $\beta$-alanine molecule attached ($m/z = 250$), shown in blue boxes, with water release. Panels (**a–d**) represent four possible pathways starting with different configurations of the protonated trimer. Relative energies (in eV) computed at the CCSD/6-311++G(d,p) level over the geometry optimized with the AM1 method and referred to the most stable neutral trimer are given next to the molecular structures. In square brackets: relative energy computed with DFT-M062X/6-311++G(d,p). H atoms are given in white, C atoms in gray, N atoms in blue, and O atoms in red.

correlation between the fragment $m/z = 45$ and fragments with masses $m/z = 72$ and $m/z = 134$ agrees with the predictions shown in Fig. 3 for the fragmentation of doubly charged dimers. In addition, the dipeptide ($m/z = 161$) is also observed with fragments $m/z = 45$ requiring the decay of larger clusters. Similarly, the coincidence with the protonated monomer ($m/z = 90$) may be explained by the decay of larger doubly charged clusters, which first emit a protonated $\beta$-alanine molecule, before the other charged residue breaks apart releasing a $m/z = 45$ unit.

Finally, we show in Fig. 5 the critical steps for dipeptide formation starting from different weakly bound trimers, protonated at the amino group ($m/z = 268$). These reactions are also explained by molecular dynamics calculations (see Supplementary Table 1). All the reactions shown in Fig. 5 correspond to concerted mechanisms that imply the emission of an $H_2O$ molecule through transition states with barriers of ~2–4 eV, leading to the formation of a protonated dipeptide ($m/z = 161$) attached to a loosely bound nonprotonated $\beta$-alanine molecule, with the exception of the lower right configuration giving a nonprotonated dipeptide loosely bound to a protonated monomer (see Fig. 5d). Similar reactions for tripeptide formation

starting from protonated dipeptide and a loosely bound $\beta$-alanine molecule have been also explored (see Supplementary Fig. 4). Furthermore, alternative mechanisms for peptide bond formation starting from the nonprotonated $\beta$-alanine dimer have been studied (see Supplementary Fig. 5). In experiments, tetrapeptides can also be easily formed as shown in Supplementary Fig. 1.

By combining state-of-the-art experimental and theoretical approaches, we have unambiguously shown the formation of peptide bonds in collisions of a single He²⁺ ion with $\beta$-alanine clusters. We attribute the formation of polypeptides to specific energy transfers resulting from the collision with He²⁺ ions, which are in contrast to photon collisions not localized but rather distributed along the ion trajectory in the cluster. Excitation and ionization processes in the collision are followed by proton transfer that leads to weakly bound protonated molecular clusters. They are stabilized through the formation of peptide bonds, releasing water molecules, via low-energy barriers.

## Methods

**Experiment.** The experiments were performed at the low-energy ion beam infrastructure ARIBE of the GANIL facility in Caen, France[35]. A bunched beam of 30 keV

He$^{2+}$ ions is produced in an electron cyclotron resonance ion source, providing ion pulses with 0.5 μs in length and with a repetition frequency of 5 kHz. In a crossed-beam experiment[36], the He$^{2+}$ beam interacts with an effusive beam of neutral clusters. The clusters are formed in a gas aggregation cluster source, where the vapor of β-alanine produced by evaporation of a powder in an oven device, aggregates in a liquid-nitrogen-cooled He buffer gas. The reaction products are mass-over-charge analyzed in a linear time-of-flight spectrometer[37]. Their number $k$ and their time-of-flight values are registered and stored in an acquisition system for each ion pulse in an event-by-event mode[28]. For very low count rates (≪1 per ion pulse), the number $k$ corresponds to the number of charged fragments produced in one collision, and hence to the charge of the fragmenting parent cluster ion. Due to the data storing, mass spectra for collision events with a given number $k$, so-called $k$-stop mass spectra, can be constructed. For more detailed experimental information, see the experimental section given in the Supplementary Information.

**Theory**. MD calculations were carried out by using TURBOMOLE[38] and the geometries and energies of the critical points in the potential energy surface (PES) by using the Gaussian09[39] program packages. MD simulations were performed in the framework of the Born–Oppenheimer Molecular Dynamics (BOMD) method, assuming classical motion of the nuclei in the electronic potential computed with DFT, in particular with the M062X functional[40]. The same functional was employed to obtain the geometries of the PES critical points. Accurate values of the energies (in the most relevant critical points of the PES) were also computed with the high-level ab initio method CCSD (Coupled Cluster method, including single and double excitations).

## Data availability

The data supporting this study are available from the corresponding author upon reasonable request. Source data are provided with this paper.

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

## Acknowledgements

The experiments have been performed at the ARIBE facility, part of large-scale infrastructure GANIL. The research was conducted in the framework of the International Associated Laboratory (LIA) Fragmentation DYNAmics of complex MOlecular systems —DYNAMO, funded by the Centre National de la Recherche Scientifique (CNRS) and in the frame of the COST action CA18212 Molecular Dynamics in the GAS phase (MD-GAS). We further acknowledge the technical support by J.-M. Ramillon, C. Feierstein, and T. Been, and the contributions of B. Manil, V. Bernigaud, T. Schlathölter, R. Hoekstra, and S. Bari in earlier studies. The authors acknowledge the generous allocation of computer time at the Centro de Computación Científica at the Universidad Autónoma de Madrid (CCC-UAM). This work was partially supported by the MICINN—Spanish Ministry of Science and Innovation—projects FIS2016-77889-R and CTQ2016-76061-P, "Severo Ochoa" Programme for Centres of Excellence in R & D (SEV-2016-0686) and "María de Maeztu" Programme for Units of Excellence in R & D (CEX2018-000805-M).

## Author contributions

P.R., M.C., and B.A.H. conceived the experimental study. P.R., M.C., A.D., L.A., and B.A.H. performed the experiments. D.G.P., F.M., M.A., and S.D.T. conceived the theoretical study. D.G.P. and S.D.T. performed the calculations. P.R. and B.A.H. wrote the initial version of the paper. All authors contributed to the interpretation of the data, as well as edited and approved the final version of the paper.

## Competing interests

The authors declare no competing interests.
