## [Peer Review File · Nature Communications]

Reviewers' comments:

Reviewer #1 (Remarks to the Author):

In this work Rousseau and co-workers study the formation of peptide bonds from clusters of β -alanine induced by the impact of He^{2+} particles. Experiments are accompanied by simulations.

The topic is relatively broad and can be of interest for the readership, however there are some key points that must be clarified.

In particular, the results (and use) of "ab initio" molecular dynamics present some unclear aspect. The most striking one is on the connection between these simulations and the important result (formation of peptide bond). Results of MD are basically shown only in Table S1:

1. I cannot see any trajectory showing H_2O loss and thus the formation of peptide bond! So how the connection between these results and experiments is possible?
2. The PES presented in the article show such reactions, but finally the authors say that they come from MD results. How this is possible if what is shown from MD does not present any of these products??
3. From what I can argue from computational details, the authors did one trajectory per each value of n (cluster size), q (charge) and excess energy. This is surely not statistically representative of the reactivity.

The authors must explain better the MD results, showing more details and more in general how they can claim things that are not evident from the manuscript and SI. For example, experiments and PES show and describe the reaction $179 \rightarrow 161 + 18$, but this is not reported in Table S1 (so in MD simulations). The same for reactions shown in Figure 5: in table S1 for $n=3$ and $q=1$ one can see m/z 84 but no H_2O loss.

As results are shown, I cannot say that conclusions are fully supported by the MD simulations, while PES and experimental results are nice and interesting.

Minor remarks:

- a) More details on experiments can be added in the SI instead of totally refer to Ref. 34.

b) I do not see why the introduction starts with a technological motivation, while the motivation of the study is related to prebiotic chemistry (as stressed in the abstract and all the paper). Authors can consider a re-organization of the introduction.

c) A reader not expert in such gas phase experiments can be confused about n-stop spectra discussion: please try to explain for a general readership.

d) In Figure 5 it can be useful to add m/z as in other similar figures.

e) The difference between reactions with $E < 0$ (as for $q=1$) and $E > 0$ ($q=2$) merits a deeper discussion on the consequence of it on the reactivity in prebiotic conditions.

f) Authors call simulations “ab initio” MD but finally they used DFT, I feel that using the word “DFT-based” MD is more honest. I know that some authors use the AIMD name when using DFT, but (i) normally in the context of Car-Parrinello simulations; (ii) I think that also in this case it is misleading for a reader which does not know the details. This is clearly a minor point but the modification is simple.

g) I am not convinced that a time step of 1 fs is correct to describe chemical reactions. Usually, smaller values (0.1 – 0.2 fs) are necessary for a correct integration of equations of motion. Please specify also which algorithm was used.

h) Which algorithm was used to randomly distribute the excess energy on the vibrational mode? Please specify.

Reviewer #2 (Remarks to the Author):

The authors have shown through a combination of experiment and theory that peptide bond formation occurs on collision of 30 keV alpha particles with amino acid cluster of β -alanine. This offers a gas-phase route to formation of small (and possibly larger) peptides in interstellar gas clouds and other environments. Convincing evidence of peptide bond formation is provided, and I believe the work is worthy of publication in Nature Communications. However, there are a number of aspects of the work that merit further discussion in the text.

1. As far as I can tell, the authors have only considered ground-state processes in their analysis. There is no explicit discussion of the range of energies that can be transferred from the alpha particles to the clusters. However, given the 30 keV energy of the alpha particles, I think there should at least be some discussion of the importance or otherwise of possible excited-state reaction pathways.

2. I would like to see some discussion about the likely accuracy and/or uncertainties in the energies calculated via the density functional calculations. In the Methods section, it is implied that some of the stationary points are calculated at a low level of theory, and others at a higher level, but it is not entirely clear which calculations are shown in the figures. Perhaps the figure captions could be revised in order to make this clear. Many of the calculated structures are relatively close in energy, so it would be very helpful to know how much faith we should place in these energies.

3. As currently presented, the discussion of "n-stop spectra" and coincidence/correlation measurements will unfortunately be relatively meaningless to anyone who is not intimately familiar with the experimental techniques employed by the authors. I suspect this is likely to be the majority of readers in a journal aimed at a fairly general scientific audience. This section should be rewritten in order to provide sufficient explanation for the intended audience to understand the arguments.

4. The characteristics of the alpha particle beam and cluster beam should be stated in the 'Experimental details' section. Also, in this section the authors refer to an energy of "-19 keV". The minus sign should be deleted.

I also have a number of minor points that the authors might like to consider.

1. Given that the entire manuscript is about peptide bond formation in amino acid clusters, starting the introduction with something of a 'red herring' about technological polymers seems rather odd. I would suggest simply deleting this sentence.

2. Throughout the manuscript there are a number of italicised subscripts that should be roman (e.g. E_{exc}). They are correct in the Supplementary Information.

3. In the Supplementary Information, the table captions should come before the tables rather than after.

Reviewer #1:

Comment:

*In this work Rousseau and co-workers study the formation of peptide bonds from clusters of β -alanine induced by the impact of He^{2+} particles. Experiments are accompanied by simulations. **The topic is relatively broad and can be of interest for the readership**, however there are some key points that must be clarified.*

Response:

We thank the referee for his/her positive evaluation of our work. We have considered all his/her suggestions.

Comment:

In particular, the results (and use) of "ab initio" molecular dynamics present some unclear aspect. The most striking one is on the connection between these simulations and the important result (formation of peptide bond). Results of MD are basically shown only in Table S1:

Response:

We thank the reviewer for pointing out this aspect. We agree with the reviewer and in order to clarify the direct connection between molecular dynamics, potential energy surfaces (PES) and peptide bond formation we performed additional DFT-based molecular dynamics simulations: new 200 trajectories were computed.

Firstly, we sampled (with 100 trajectories) the doubly-charged trimer, i.e., the smallest possible cluster that can lead to the protonated dimer studied in the PES shown in Figure 2. The new MD results show that 90% of the trajectories lead to the protonated singly-charged dimer, and subsequent fragmentation of the deprotonated singly-charged monomer. Both observations are clearly seen in the experiments with the detection of fragments with $m/z=179$ [$(\beta\text{-ala})_2\text{H}$]⁺ and other smaller fragments, such as those with $m/z=45$. We have added a new figure in the Supplementary Information (Fig.S2) with some examples of trajectories that represent these findings.

Secondly, we run additional 100 trajectories starting from the protonated, singly-charged dimer, in particular starting from the transition state geometry showed at relative energy -7.8(-7.9) eV (see Fig.2), which is precisely the determining step in the mechanism found for the peptide bond formation. 53% of those MD simulations followed the PES toward water removal and peptide bond formation. An additional figure was added in the Supporting Material (Fig.S3) showing some trajectories as examples of the peptide bond formation.

The theory part in the methods section has been substantially reduced (lines 236-245), including only the most important aspects of the calculations and moving all the details to the supplementary information.

Comment:

1. *I cannot see any trajectory showing H₂O loss and thus the formation of peptide bond! So how the connection between these results and experiments is possible?*

Response:

In the new MD simulations, we observe water emission accompanied by peptide bond formation (see above). These results are explained in the amended manuscript and detailed in the Supplementary Information (Fig. S3).

Comment:

2. *The PES presented in the article show such reactions, but finally the authors say that they come from MD results. How this is possible if what is shown from MD does not present any of these products??*

Response:

In the new MD simulations, we observe that one of the main products is the protonated dimer, the starting point to explore the rest of the potential energy surface (see structure $[(\beta\text{-ala})_2+\text{H}]^+$ Fig.2). In the amended manuscript we explain these results, providing figures with examples of trajectories in the Supplementary Information (Fig. S2).

Comment:

3. *From what I can argue from computational details, the authors did one trajectory per each value of n (cluster size), q (charge) and excess energy. This is surely not statistically representative of the reactivity.*

Response:

We agree with the reviewer that this is surely not a statistical representation of the aforementioned reactivity. The purpose of these simulations was to provide the starting point for the PES construction and briefly check the effect of the cluster size, charge and excess energy on the intermolecular hydrogen transfer. The latter processes occur at different timescales, depending on the cluster size and is the prominent channel, as observed in the experiments.

We have extended the MD simulations to clarify this point. We have considered 200 trajectories as detailed above. The statistical analysis on these trajectories are given in Figs. S2 and S3 of the revised version of the Supplementary Information.

Comment:

The authors must explain better the MD results, showing more details and more in general how they can claim things that are not evident from the manuscript and SI. For example, experiments and PES show and describe the reaction $179 \rightarrow 161 + 18$, but this is not reported in Table S1 (so in MD simulations). The same for reactions shown in Figure 5: in table S1 for $n=3$ and $q=1$ one can see m/z 84 but no H_2O loss. As results are shown, I cannot say that conclusions are fully supported by the MD simulations, while PES and experimental results are nice and interesting.

Response:

Following the reviewer's suggestion, a detailed statistical analysis of the new MD calculations described above has been included in the Supplementary Information. Although the results of these simulations agree with the experimental observations, and support our previous interpretation based on the analysis of the PES, we decided to move the discussion to the Supplementary Information. We thus highlight the PES results and the experiments in the main article.

Comments (minor remarks):

Minor remarks:

a) More details on experiments can be added in the SI instead of totally refer to Ref.34.

Response:

A short description of the experiment is now given in the manuscript (line 220-235). A more extended section, specifying much more details of the experiment

and the experimental procedure is given in the SI, as requested by both referees.

Comment:

b) I do not see why the introduction starts with a technological motivation, while the motivation of the study is related to prebiotic chemistry (as stressed in the abstract and all the paper). Authors can consider a re-organization of the introduction.

Response:

We have removed the sentence, and reorganized the introduction as suggested by the reviewer.

Comment:

c) A reader not expert in such gas phase experiments can be confused about n -stop spectra discussion: please try to explain for a general readership.

Response:

We provide a wider explanation to clarify this point in the text (lines 176-181) and in the experimental sections.

To avoid confusion with n the size of the cluster, we also change the notation n -stop to k -stop.

Comment:

d) In Figure 5 it can be useful to add m/z as in other similar figures.

Response:

We have included m/z in Fig.5.

Comment:

e) The difference between reactions with $E < 0$ (as for $q=1$) and $E > 0$ ($q=2$) merits a deeper discussion on the consequence of it on the reactivity in prebiotic conditions.

Response:

We thank the reviewer to point out this important finding. Indeed, proton transfer facilitates the reactivity since the protonated dimer is more stable than the most stable neutral structure (see the potential energy surface). i.e., proton transfer stabilizes the dimer and there is no need for extra energy to be transferred in the collision to induce the formation of the peptide. In contrast, the analysis of the potential energy surface for the doubly-charged system shows that reactivity requires additional energy (which can be obtained from the collision) to overcome the barriers that connect the neutral structure with the reaction products. We have followed the referee's recommendation and we have extended the discussion about this point in the amended manuscript (lines 116-130).

Comment:

f) Authors call simulations "ab initio" MD but finally they used DFT, I feel that using the word "DFT-based" MD is more honest. I know that some authors use the AIMD name when using DFT, but (i) normally in the context of Car-Parrinello simulations; (ii) I think that also in this case it is misleading for a reader which does not know the details. This is clearly a minor point but the modification is simple.

Response:

The reviewer is right. In our simulations the nuclear motion is classical and is calculated on the fly by using the forces associated with the DFT energy. The term DFT-based MD is more appropriate. We have accordingly changed it in the amended manuscript.

Comment:

g) *I am not convinced that a time step of 1 fs is correct to describe chemical reactions. Usually, smaller values (0.1 – 0.2 fs) are necessary for a correct integration of equations of motion. Please specify also which algorithm was used.*

Response:

The new simulations have been performed with a more accurate time step of 0.1 fs as suggested by the reviewer. We use the Leapfrog Verlet algorithm to integrate the equations of motion. We have added a sentence to clarify this point in the computational details given in the revised version of the Supplementary Information.

Comment:

h) *Which algorithm was used to randomly distribute the excess energy on the vibrational mode? Please specify.*

Response:

The excess of energy was randomly distributed in the nuclear degrees of freedom, not projected on the vibrational modes. We have clarified this point in the computational details given in the revised version of the Supplementary Information.

Reviewer #2:

Comment:

*The authors have shown through a combination of experiment and theory that peptide bond formation occurs on collision of 30 keV alpha particles with amino acid cluster of β -alanine. This offers a gas-phase route to formation of small (and possibly larger) peptides in interstellar gas clouds and other environments. Convincing evidence of peptide bond formation is provided, and **I believe the work is worthy of publication in Nature Communications**. However, there are a number of aspects of the work that merit further discussion in the text.*

Response:

We thank the referee for his/her positive evaluation of our work. We have considered all his/her suggestions.

Comment:

1. *As far as I can tell, the authors have only considered ground-state processes in their analysis. There is no explicit discussion of the range of energies that can be transferred from the alpha particles to the clusters. However, given the 30 keV energy of the alpha particles, I think there should at least be some discussion of the importance or otherwise of possible excited-state reaction pathways.*

Response:

We agree with the referee. Most probably excited states are populated in the collision with the alpha particles, since electron capture can occur from both outer and inner valence orbitals of the cluster, thus leading to the formation of ions in excited states. Theoretical evaluation of the excited-state populations would require to describe the dynamics of the collision process, which is not possible by using existing methods for ion-molecule collisions due to the extreme complexity of the target (existing methods only allow one to describe collisions involving small molecular targets, typically containing one to five atoms). In addition, even if we assumed a given "ad hoc" population of some excited states of the cations after the collision, performing accurate molecular dynamics simulations for molecules not initially in their ground state require an enormous amount of computer time. Due to the large size of the generated cations, only TDDFT-MD would be affordable, but, as the existing literature shows, very few trajectories can be considered (usually one or two for systems as large as the ones investigated in this work!), so that the results will not have any statistical meaning.

Our approach, which assumes that the generated cations are in the ground state but are hot is based on the fact that (i) the collision is much faster than fragmentation (the typical collision time is of the order of the femtosecond) and (ii) the energy available in the excited electronic states is rapidly redistributed into the nuclear degrees of freedom due to the very efficient non-adiabatic couplings (e.g., through conical intersections) between these electronic states and the very dense manifold of vibrational states associated with such a large system. In this way, introduction of the excitation energy in the form of nuclear velocities to start the fragmentation dynamics is not only the only viable way to perform calculations but it is also a reasonable assumption. This approach has been shown to accurately describe fragmentation dynamics in similar experiments in the past (Ref. 28-32 of the revised manuscript). As this is a very important point, we have included this discussion in the revised version of manuscript (lines 90-99), so that the reader can better understand the limitations of our method but also

why these limitations should not be a serious problem for such a large system when fragmentation takes from several hundred femtoseconds to picoseconds.

Comment:

2. *I would like to see some discussion about the likely accuracy and/or uncertainties in the energies calculated via the density functional calculations. In the Methods section, it is implied that some of the stationary points are calculated at a low level of theory, and others at a higher level, but it is not entirely clear which calculations are shown in the figures. Perhaps the figure captions could be revised in order to make this clear. Many of the calculated structures are relatively close in energy, so it would be very helpful to know how much faith we should place in these energies.*

Response:

We thank the reviewer for pointing out this important aspect. We run additional calculations using a high level of ab initio theory, namely the Coupled Cluster method including single and double excitations in combination with a triple-z basis set including extra diffuse and polarization functions: CCSD/6-311++G(d,p). With this level of theory, we are sure that the relative energies are computed with good accuracy. These simulations have been performed for the most relevant points in the potential energy surface. In particular all critical points in Figures 2 and 3. In these figures DFT and CCSD relative energies are given for comparison. DFT provides very similar results to those computed at the CCSD level.

Accordingly, we corrected the corresponding captions of the figures and added a sentence comparing both methods (lines 105-108). We have also changed the numbers in the figures and in the text.

The theory part in the methods section has been substantially reduced (lines 236-245), including only the most important aspects of the calculations and moving all the details to the supplementary information.

Comment:

3. *As currently presented, the discussion of "n-stop spectra" and coincidence/correlation measurements will unfortunately be relatively meaningless to anyone who is not intimately familiar with the experimental techniques employed by the authors. I suspect this is likely to be the majority of readers in a journal aimed at a fairly general scientific audience. This section should be rewritten in order to provide sufficient explanation for the intended audience to understand the arguments.*

Response:

We agree with the reviewer. Probably we did not provide enough explanations for the broad audience of the journal. We have extended the discussion of the n-stop spectra (lines 176-181) and clarified the key aspects, in particular in the SI.

To avoid confusion with n the size of the cluster, we also change the notation n -stop to k -stop.

Comment:

4. *The characteristics of the alpha particle beam and cluster beam should be stated in the 'Experimental details' section. Also, in this section the authors refer to an energy of "-19 keV". The minus sign should be deleted.*

Response:

We have included the experimental details in the SI as requested by the reviewer and we have corrected the typos.

Comments (minor remarks):

I also have a number of minor points that the authors might like to consider.

Comment:

1. Given that the entire manuscript is about peptide bond formation in amino acid clusters, starting the introduction with something of a 'red herring' about technological polymers seems rather odd. I would suggest simply deleting this sentence.

Response:

We have removed the sentence, and reorganized the introduction as suggested by the reviewer.

Comment:

2. Throughout the manuscript there are a number of italicised subscripts that should be roman (e.g. E_{exc}). They are correct in the Supplementary Information.

Response:

We have changed this formatting typo.

Comment:

3. In the Supplementary Information, the table captions should come before the tables rather than after.

Response:

We have changed the place of the table captions in the Suppl. Info.

REVIEWERS' COMMENTS:

Reviewer #1 (Remarks to the Author):

The authors modified the manuscript largely, in particular performing additional MD simulations, so now the manuscript is more consistent and it can be published.

Reviewer #2 (Remarks to the Author):

In revising their manuscript, the authors have addressed all of the points I raised in my original review with some aplomb! They are to be congratulated on their thoroughness. I am satisfied with the manuscript in its revised form and I recommend publication with no further changes.

Claire Vallance